# Genome Sequencing and Analysis of *Nigrospora oryzae*, a Rice Leaf Disease Fungus

**DOI:** 10.3390/jof10020100

**Published:** 2024-01-26

**Authors:** Qian Zhao, Liyan Zhang, Jianzhong Wu

**Affiliations:** 1Cultivation and Farming Research Institute, Heilongjiang Academy of Agricultural Sciences, Harbin 150086, China; zhaoqian0401@sina.com; 2Forestry College, Inner Mongolia Agricultural University, Huhhot 010011, China; zhangliyanyong@126.com; 3Institute of Forage and Grassland Sciences, Heilongjiang Academy of Agricultural Sciences, Harbin 150086, China

**Keywords:** *Nigrospora oryzae*, fungal pathogens, genome, transcription factors

## Abstract

*Nigrospora oryzae* is one of several fungal pathogens known to cause brown streaks, leaf spots, and latent infections in rice. In this study, the entire 42.09-Mb genome of *N. oryzae* was sequenced at a depth of 169× using the Oxford Nanopore Technologies platform. The draft genome sequence was comprised of 26 scaffolds, possessed an average GC content of 58.83%, and contained a total of 10,688 protein-coding genes. Analysis of the complete genome sequence revealed that CAZyme-encoding genes account for 6.11% of all identified genes and that numerous transcription factors (TFs) associated with diverse biological processes belong predominantly to Zn-clus (22.20%) and C_2_H_2_ (10.59%) fungal TF classes. In addition, genes encoding 126 transport proteins and 3307 pathogen–host interaction proteins were identified. Comparative analysis of the previously reported *N. oryzae* reference strain GZL1 genome and the genome of a representative strain ZQ1 obtained here revealed 9722 colinear genes. Collectively, these findings provide valuable insights into *N. oryzae* genetic mechanisms and phenotypic characteristics.

## 1. Introduction

*Nigrospora oryzae* (Berk. & Broome) Petch, a fungal species belonging to the kingdom *Fungi*, phylum *Ascomycota*, class *Sordariomycetes*, order *Xylariales*, family *Xylariaceae*, and genus *Nigrospora*, can infect a broad range of crops, including rice (Oryza sativa), corn, wheat, cotton, ginger, dayflower, and others [1,2,3,4,5]. *N. oryzae* also causes leaf spot or blight diseases in various non-crop species, including cotton rose (*Hibiscus mutabilis* Linn.), a deciduous shrub native to China. Infections caused by this pathogen occur with very high incidence from May to September, presenting as irregular black spots on leaves that grow and coalesce into large, black necrotic areas that progress to cause leaf chlorosis and abscission [6]. Notably, *N. oryzae* leaf spot also occurs in Asiatic dayflower, an annual herbaceous weed found throughout China, prompting researchers to investigate *N. oryzae* as a potential biological agent for use in eradicating this weed [7].

In addition to leaf spots, *N. oryzae* can also cause rice leaf brown stripe disease, a widespread disease occurring in certain areas of northeast China with increasing frequency each year. The disease primarily triggers color changes on leave surfaces that progress from green to yellow–brown and eventually to black–brown, resulting in substantial leaf damage and eventual destruction. Severe infestations can lead to panicle desiccation and the formation of poorly developed and/or weakened rice grains that can significantly reduce overall yield. Additionally, *N. oryzae* can latently colonize rice and various other plant species, enabling the spread of this pathogen into primary rice-producing areas, with potentially dire consequences.

The detrimental impact of *N. oryzae* infestations on agricultural production has been noted in the literature for over 70 years [8]. This disease not only results in reduced rice yield and diminished milling quality, but has been mistaken for rice blast, which cannot destroy all of the functions of the branches, due to similar panicle rot-related symptoms. Since 2020, this misidentification has led to unnecessary spraying of rice paddies in Hebei, Anhui, and Zhejiang provinces of China, regions which have not traditionally been affected by rice blast. Therefore, it is important to obtain the genomic information to understand the molecular mechanism of rice disease caused by *N. oryzae*.

Beyond leaf damage, *N. oryzae* infections can extend to primary or secondary branches and pedicels, resulting in the formation of brown- or black-colored lesions on stems. Unlike rice blast, these lesions do not appear on grains or necks of spikes. This distinction allows for the differentiation between *N. oryzae* infection and rice blast, as *N. oryzae* does not entirely disrupt panicle function, with blank panicles rarely observed [9]. However, *N. oryzae* infection can lead to incomplete grain filling, often resulting in grain weight loss ranging from 5% to 25% [9].

At present, there are few prevention and control methods for *N. oryzae*, which mainly focus on the use of some chemical agents and biological control measures. Fungicidal suspensions used to prevent and control *N. oryzae* primarily contain benzene ether mecycloazole, nitrile, fresh amine manganese salt, phenylpropiazole, pentyl azole alcohol water dispersion agent, and other agents. These formulations exert strong inhibitory effects on mycelial growth and spore production. Emerging fungal-control strategies include innovative approaches guided by results of dual-culture studies of the Chrysanthemum morifolium phyllosphere conducted using potato dextrose agar (PDA) plates. The results of these studies revealed antagonistic effects of 12 out of 231 bacterial strains on *N. oryzae* growth, among which Bacillus siamensis D65 exhibited greatest inhibition of *N. oryzae* growth [10]. Additionally, the novel 1,5-disubstituted-1H-pyrazole-4-carboxamide derivative Y47 has been shown to inhibit *N. oryzae* growth, as reflected by a Y47 EC50 value of 9.2 mg/L [11]. Nevertheless, due to the emergence of fungal pesticide resistance in response to the continued environmental release of large quantities of pesticide-based pollutants, there is a pressing need for safer and more effective strategies for preventing and controlling *N. oryzae*.

Although chemical and biological control measures have certain effects on agriculture, long-term use may have adverse effects on the ecological environment. For example, chemical pesticides can contaminate water, soil, and agricultural products, while also harming non-target organisms. Biological control, while relatively environmentally friendly, can also lead to disruption of the ecological balance, as introduced organisms may become new invasive species and pose a threat to the local ecosystem.

In contrast, improving disease resistance through breeding is a more sustainable and environmentally friendly approach. By breeding varieties with disease-resistant traits, the use of chemical pesticides can be reduced and the negative impact on the environment can be reduced. At the same time, the cultivation of disease-resistant varieties also helps to improve crop yield and quality and increase farmers’ income.

In view of the adverse impact of chemical and biological control measures on the agricultural ecological environment, breeders believe that breeding for disease resistance is a direct and effective measure to improve the disease resistance of crops. Therefore, exploring the pathogenic mechanism of the pathogen is an important link in opening up the field of disease-resistant biological breeding, and it is also the most popular among environmental protection personnel and agricultural producers.

The emergence of whole-genome sequencing technologies, notably the Pacific Biosciences (PacBio) SMRT and Illumina sequencing platforms, has revolutionized the generation of public whole-genome assemblies for various pathogenic fungi [12,13]. Furthermore, the availability of whole-genome sequence data has enabled extensive data mining that has led to discoveries of key pathogenic genes, such as those encoding secondary metabolites, chitin-binding proteins, plant cell wall-degrading enzymes, and expanded groups of protein-coding genes with other putative pathogenic functions [14,15,16]. Here, we present a draft genome assembly for a strain of *N. oryzae* isolated from a diseased rice plant in northeast China. This assembly serves as a foundational resource for future investigations into *N. oryzae* genomic features and pathogenic mechanisms.

## 2. Materials and Methods

### 2.1. Isolation of Strain

In September 2019, we conducted a study in three major rice producing areas in Heilongjiang Province, China. The infected leaves were collected from Harbin City (N 45°03′, E 127°03′), Jiamusi City (N 46°48′, E 130°22′), and Suihua City (N 46°38′, E 126°58′). These leaves were then purified and cultured. Based on the mycelial growth characteristics observed on PDA medium, we selected the ZQ1 strain as the representative strain due to its largest proportion of uniform morphology. Subsequent tests were conducted using this strain.

Tissue samples excised from spots on *N. oryzae*-infected rice leaves collected from the field were disinfected by immersion in 70% ethanol for 2 min, washed 3 to 4 times with sterile distilled water, then placed onto surfaces of sterile PDA plates. Plates containing tissue samples were placed in an incubator at 25 °C for 3 d. Next, a small sample of mycelium was removed from each plate and cultured on a fresh PDA plate for 3 d to allow it to grow into a colony of about 1 cm in diameter. This process was repeated 3 to 4 times to generate a pure *N. oryzae* strain.

Using Mahuka’s method [17], solid cultures of purified *N. oryzae* strain were incubated in a 30 °C incubator for 3 d, then fresh mycelia at the edge of each colony were transferred to culture bottles containing potato glucose liquid medium (50 μg per bottle). Bottles were incubated at 30 ± 1 °C while being rotated at a speed of 120 r/min. After 4 to 5 d, the fungal cultures were centrifuged (Himac CR22G, Koki Holding Co. Ltd., Tokyo, Japan) and 100 mg of each cell pellet was frozen in liquid nitrogen, ground, and processed using a fungal DNA extraction kit (50t, OMEGA Bios Inc., Los Angeles, CA, USA). All extractions were performed according to the manufacturer’s instructions provided with the kit.

### 2.2. Identification of Strain

PCR amplification was used to generate products for strain identification using ITS1 universal primers designated ITS1 (5′-TCCGTAGGTGAACCTGCGG-3′) and ITS4 (5′-TCCTCCGCTTATTGATATGC-3′). The amplification was carried out in a 50 μL reaction system, as we previously reported publicly in a description in [18].

### 2.3. Whole-Genome Sequencing

To improve the accuracy of genome sequencing, raw sequence data for the genome of the *N. oryzae* ZQ1 strain were generated based on two sequencing platforms, next-generation sequencing (NGS) and Oxford nanopore technology (ONT). After sequence data were filtered to remove adaptors, short fragments, and low-quality sequences, clean data consisting of filtered subreads were analyzed, confirming a sequencing depth of 169×. Next, the complete genome was assembled using Canu v1.5 [19] software with the set parameters Grid = false, overlapper = mhap, and utgReAlign = true. Next, wtdbg2 [20] (https://github.com/ruanjue/wtdbg2, accessed on 12 March 2022) was used to generate a highly accurate long-read genomic sequence assembly.

The completeness of the final genome assembly was assessed using the Pilon v1.22 software tool with the set parameters-mindepth 1 and -changes-fix all [21]. The NGS sequencing data were compared with the assembled genome to assess the integrity of the genome assembly to avoid reducing the alignment rate due to inter-individual sequence differences. Here, the genome assembly integrity was evaluated from the two aspects of duplication ratio and Benchmarking Universal Single-Copy Orthologs (BUSCO) using BWA v0.7.9-1 software [22] and BUSCO v2.0 [23] with the default parameters, respectively.

### 2.4. Genomic Component Analysis

Due to the relatively low conservation of interspecies repeat sequences, a specific repeat sequence database needs to be constructed when predicting repeat sequences for specific strains. Both structure prediction and de novo prediction methods were utilized; we employed a comprehensive sequence prediction approach incorporating the use of software tools LTR_FINDER v1.05 [24], MITE-Hunter v2 [25], RepeatScout v1.0.5 [26], and PILER-DF v2.4 [27] to construct the repetitive sequence database of the *N. oryzae* genome. Thereafter, PASTEClassifier (pseudo agent system for transposable elements classification) [28] was used to classify transposable elements and repetitive sequences to generate a database that was subsequently combined with the Repbase database [29] to create the final repeat sequence database utilized here. Finally, RepeatMasker v4.0.6 [30] software was used to predict fungal repeat sequences within the *N. oryzae* genome using the final repeat sequence database described above.

### 2.5. Protein-Encoding Gene Prediction

Gene structure prediction was conducted based on results of de novo prediction, homologous protein prediction, and transcriptomic evidence. For de novo gene structure prediction, we used software tools Genscan v1.0 [31], Augustus v2.4 [32], GlimmerHMM v3.0.4 [33], GeneID v1.4 [34], and SNAP (version 2006-07-28) [35]; for homologous protein prediction, we used GeMoMa v1.3.1 [36]; for transcript assembly based on reference transcripts, we used Hisat2 v2.0.4 [37] and StringTie v1.2.3 [34]. To predict unigene sequences based on transcriptomic evidence, we used TransDecoder v2.0 [38] and PASA v2.0.2 [39]. After completion of the abovementioned analyses, Ensemble Variant Mapping software (EVM v1.1.1) [40] was used to integrate the abovementioned prediction results, then PASA v2.0.2 was used to annotate each identified protein-coding sequence (CDS).

### 2.6. Gene Function Annotation

Gene function annotation was conducted by aligning predicted protein-encoding gene sequences against sequences within the following databases: Nr (Non-Redundant Protein Sequence Database) (https://www.ncbi.nlm.nih.gov/refseq/, accessed on 4 July 2022), COG (https://www.ncbi.nlm.nih.gov/COG/, accessed on 5 July 2022), KEGG (https://www.kegg.jp/kegg/kegg1.html, accessed on 5 July 2022), and Swiss-Prot (https://www.uniprot.org/, accessed on 6 July 2022). Functional annotation of genes was carried out using Blast2GO (https://www.blast2go.com/, accessed on 3 May 2022) [41], leveraging the Nr database alignment to assign gene ontology (GO) terms from the GO knowledgebase [42]. Pfam functional annotation was performed using HMMER (http://hmmer.org/download.html, accessed on 4 May 2022) [43] with the Pfam database [44]. Further gene functional annotation analysis encompassed COG and KEGG metabolic pathway enrichment analysis and GO functional enrichment analysis.

### 2.7. Data Availability

The *N. oryzae* genome sequence was deposited at NCBI under the final accession number of PRJNA827930. Filtering of the raw data to remove adapters, low-quality sequences, short fragments, and reads <2000 bp generated a clean dataset to be used for subsequent analyses (Appendix A).

### 2.8. Collinearity Analysis

In this study, collinearity analysis was primarily conducted using MCScanX v3.1 software with default parameters. This approach enabled the identification of genomic variations between the two strains, reflecting evolutionary structural changes and altered genome arrangements. Additionally, the results of this analysis provided insights into similarities between the *N. oryzae* reference strain GZL1 and the ZQ1 strain, thereby enhancing our understanding of their genetic relatedness and shared characteristics.

## 3. Results

### 3.1. Strain Identification

Growth rates of the strain ranged from 19.3 to 24.7 mm/d. During the initial growth stage, colonies were white, and then became gray after 5 days of growth. Eventually, mycelia of plates gradually turned black (Appendix A). Growth of the strain led to the generation of concentric round colonies with morphologically irregular edges that were consistent with known morphological characteristics of *N. oryzae*. To further identify the strain obtained in this study, internal transcribed spacer (ITS) sequences of the representative strain was amplified and sequenced using the primer pair ITS1/ITS4. Subsequently, we conducted sequence analysis via BLASTn on the NCBI website, revealing similarity values of the strain and *N. oryzae* of 99%. Taken together, the abovementioned morphological and molecular findings indicated that the strain was *N. oryzae*.

### 3.2. Genome Assembly and Evaluation

A comprehensive dataset was produced through nanopore sequencing that consisted of 7.73 Gb of raw data. After removing multiple filter connectors, short segments, and low-quality data, about 7.13 Gb clean data were obtained. A detailed overview of genome-related metrics is outlined in Appendix A; the clean reads’ length primarily ranged between 10,000 and 20,000 bp, accounting for 42.33% of the dataset. The complementary graphical representations presented in Appendix A depict the correlation between length and quantity of clean reads. The dataset, which exhibited a substantial sequencing depth of approximately 169×, was assembled to create a genomic sequence of length 42.09 Mb, with an overall GC content of 58.83% (Table 1, Figure 1).

Assessment of genome assembly completeness was based on two criteria. First, the Benchmarking Universal Single-Copy Orthologs (BUSCO) evaluation revealed 287 complete BUSCO genes, signifying genome completeness of 98.97%. These metrics, detailed in Appendix A, confirmed the integrity of the resulting genome assembly. Second, an evaluation of next-generation sequence data recovery provided an alignment ratio of 95.99%, indicating 99.89% sequence coverage of the assembled genome and a final sequencing depth of approximately 69× (Appendix A). Therefore, the combined sequencing technology use of NGS and ONT ensures the accuracy of strain genomes.

### 3.3. Repeat Sequence Prediction

In this study, we constructed a repeat sequence database for use in predicting repeat sequences which enabled the identification of 1,233,251 bp repeat sequences in the ZQ1 genome comprising 2.93% of clean reads. Two classes of repeat sequences were obtained, Class I and Class II, accounting for 1.90% and 0.59% of the entire genome, respectively. The highest repeat content was obtained for the Class I long terminal repeat retrotransposon (long terminal repeat, LTR), with Gypsy repeat sequences (0.9%) ranking second in abundance, followed by Copia repeat sequences (0.25%). In addition, Class II/TIR (0.57%), LINE (0.48%), and Class I/PLE/LARD (0.26%) repeats were also identified (Figure 2).

### 3.4. Protein-Coding Gene, Non-Coding RNA, and Pseudogene Prediction

Using a tripartite gene structure prediction method integrating de novo prediction, homologous protein prediction, and transcriptomic evidence, we identified a total of 10,688 protein-coding genes (Appendix A). The average length of these genes was 2356.59 bp; the average number of exons per gene was 2.96; the average CDS number per gene was 2.89; and the average number of introns per gene was 1.96 (Appendix A). Of the total 10,688 genes that were predicted through EVM integration of all three prediction methods, 10,238 (95.78%) were supported by orthologous prediction results and transcriptomic evidence, thereby confirming the high predictive power of our method (Appendix A). Ultimately, this set of genes included 53 rRNA genes belonging to 3 families, 272 tRNA genes belonging to 45 families, and 40 other ncRNAs belonging to 33 families (Figure 1). Searching for the genome in homologous sequences revealed 17 pseudogenes containing premature stop codons and frameshift mutations.

### 3.5. Transcription Factor Prediction

Transcription factors (TFs), which are essential for modulating diverse biological processes, regulate gene expression and play central roles in the development and evolution of organisms [45]. Ultimately, the functional annotation approach used in this study enabled the identification of 491 TF-encoding genes, transcription regulators, and protein kinases within the *N. oryzae* genome, accounting for 4.62% of all predicted genes. As for other fungi, genes encoding TFs (245 members) and transcription regulatory proteins (139 members) represented the two largest classes of TFs in *N. oryzae*, accounting for approximately 49.9% and 28.3% of predicted TFs, respectively, followed by Zn-clus finger TFs (109 members) and C2H2-type zinc finger TFs (52 members), which, respectively, accounted for approximately 22.20% and 10.59% of predicted TFs (Figure 3).

### 3.6. Functional Annotation of the Genome

Outbreaks of *N. oryzae*, an emerging leaf disease of rice, have increased both in number and geographic distribution in recent years. To uncover pathogenic molecular regulatory mechanisms associated with fungal growth and development, high-throughput whole-genome sequencing and assembly of the *N. oryzae* genome were performed, resulting in the identification of 10,268 putative protein-encoding genes. Functional annotation analysis of these genes conducted based on alignments of predicted protein sequences with protein sequences within KOG, KEGG, Swiss-Prot, TrEMBL, Nr, and other databases identified approximately 22.0% (2259), 13.06% (1341), and 13.17% (1352) of *N. oryzae* metabolic process-related genes derived from GO and KEGG databases, respectively (Appendix A; Appendix A).

Further analysis of the *N. oryzae* genome sequence led to the prediction of 126 and 3307 secreted proteins based on alignments with proteins within two proprietary databases, the Transporter Classification Database (TCDB) and the Pathogen–Host Interactions database (PHI), respectively. Our analysis also extended to the prediction of genes encoding carbohydrate-active enzymes (CAZymes), which have been linked to pathogenicity due to their key roles in interactions between pathogenic fungi and their hosts [6,7]. In this study, we identified genes within the draft *N. oryzae* genome encoding 653 CAZymes, including 282 glycoside hydrolases (GH, 39.11%), 127 auxiliary active family proteins (AA, 17.61%), 126 carbohydrate esterases (CE, 17.47%), 99 glycosyl transferases (GT,13.73%), 78 carbohydrate-binding modules (CBM, 10.81%), and 9 polysaccharide lyases (PSL, 1.24%).

Additionally, the *N. oryzae* genome was found to harbor 1220 predicted signal peptides and 2435 transmembrane and secreted proteins. These secreted proteins potentially function as effectors that suppress plant defense responses, thereby facilitating *N. oryzae* infection of specific plant hosts. Further analysis of these secreted proteins led to the identification of 130 proteins with putative roles in plant–pathogen interactions, while BLAST searches of these protein sequences against the Nr database revealed putative evolutionary relationships between *N. oryzae* proteins and analogous species (Appendix A). Collectively, these findings offer novel insights into molecular mechanisms related to the interplay between phenotypic traits and genetic mechanisms in *N. oryzae*.

### 3.7. Collinearity Analysis

Collinearity analysis of genome sequences for the *N. oryzae* reference strain GZL1 (a field strain) and the representative ZQ1 strain was conducted using the MCScanX toolkit, leading to the identification of 9561 shared gene families (Table 2) and 9722 collinear genes (Figure 4). Collinearity blocks were distributed among 23 scaffolds, with the greatest number of colinear genes found on scaffold 1 (1143 genes), while colinear gene numbers below 100 were associated with 5 scaffolds, and colinear gene numbers ranging between 120 and 1143 were associated with the remaining scaffolds (Figure 4, Appendix A).

## 4. Discussion

*N. oryzae* is an important phytopathogenic fungus with a broad host range encompassing rice [9], Hibiscus mutabilis [6], Asiatic dayflower [7], ginger [46], and other plants. Signs of *N. oryzae* infection of rice leaves mainly appear during middle- and late-stage growth, resulting in the formation of leaf spots and serious seed head damage leading to reduced seed setting rate. As early as 1957, Japanese scholars reported *N. oryzae*-induced rice leaf spot disease in Tokyo that could be transmitted by seeds, wind, and rain. Such infestations, which have become increasingly serious, particularly in southern China, have decimated major Chinese rice-producing areas and incurred huge economic losses.

The first reported case of *N. oryzae*-induced rice panicle branch rot disease, which occurred in a patty field in Zhejiang, China, exhibited symptoms that could easily be mistaken for symptoms associated with rice blast. This infestation not only caused substantial losses of crop yield and lower milling quality, but its resemblance to rice blast led to unnecessary spraying of rice fields with fungicides [9]. Since that time, few cases of serious *N. oryzae*-induced rice leaf brown stripe disease have been reported. The recent expansion of rice cultivation in northeast China has spurred the growth of local rice farms cultivating unique rice varieties. However, this expansion has brought with it the potential for virulent strains of pathogenic fungi and bacteria. These pathogens pose a threat to seedling health, potentially leading to diminished rice yield and quality when rice is cultivated in environments conducive to pathogen growth.

Despite inherent variations, genomes of members of a given species distributed across different regions retain significant ancestral information, possess a high level of conservation, and contain genes belonging to numerous gene families. Our comparison of *N. oryzae* strains, GZL1 and ZQ1, which were annotated using the same methodology, confirmed the presence of rich orthologous content (Figure 4). Collinearity analyses revealed both substantial conservation and sequence variations between these genomes, thereby refining our genome annotations. Moreover, the comparison unveiled extensive colinear regions containing abundant homologous data that enhanced genome annotation accuracy and uncovered unknown genes with potential roles in *N. oryzae* pathogenesis.

The development of genome sequencing technology has had a profound impact on life science research, disease diagnosis, and other fields. Genome sequencing of *N. oryzae* can reveal the genetic variation and help us understand the life process and function. By comparing the results of the ZQ1 genome with the reference genome of GZL1, the genes related to the disease were found in this study, providing more accurate diagnosis and treatment of the disease. Although great progress has been made in genome sequencing technology, there are still many challenges and problems, such as data processing, analysis, and interpretation, which need to be further studied and solved in the future.

## Figures and Tables

**Figure 1 jof-10-00100-f001:**
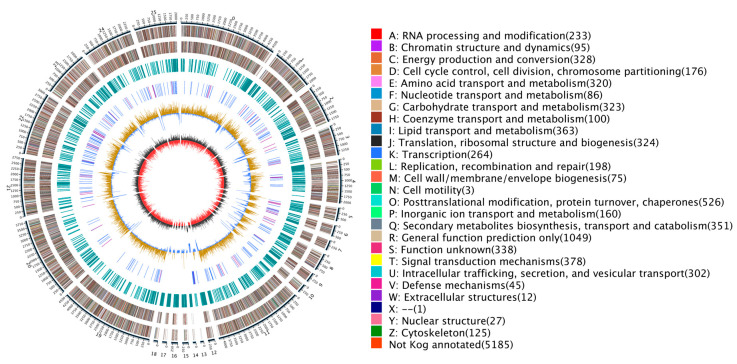
Genomic map of *N. oryzae*. The outermost circle is a marker of genome size, with each demarcation representing 5 kb, that shows the entire genome size of *N. oryzae* ZQ1 of 42.09 Mb. The second and third circles from the outside show positions of genes coded by plus and minus strands of the genome, respectively. The function and gene number annotated by COG are listed on the right of the figure whereby each function corresponds to a color and each color corresponds to a different COG functional classification. The fourth circle shows repeating sequences. The fifth circle shows genes coding for tRNA and rRNA, where blue and purple represent tRNA and rRNA, respectively. The sixth circle shows the GC content; the light yellow part shows that GC content is higher than the mean GC content of genome—the higher the peak value is, the bigger the difference is—and the blue part shows that GC content is lower than the mean GC content of genome. The innermost circle shows the GC-skew, with dark gray representing regions where G is greater than C and red representing regions where C is greater than G. Circos (https://circos.ca/, accessed on 21 January 2023) was used as the drawing program.

**Figure 2 jof-10-00100-f002:**
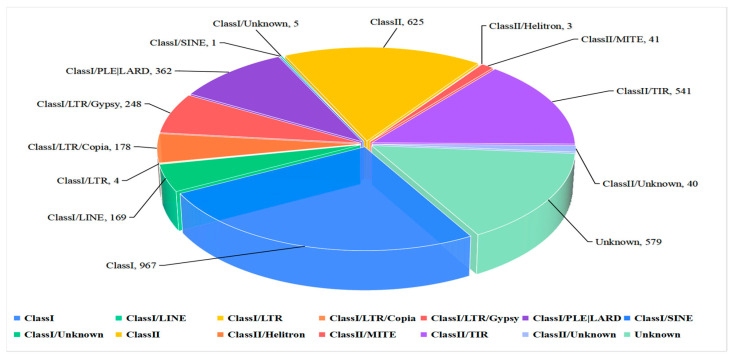
Statistical analysis and the percentage of different types of repeat sequences in the *Nigrospora oryzae* genome.

**Figure 3 jof-10-00100-f003:**
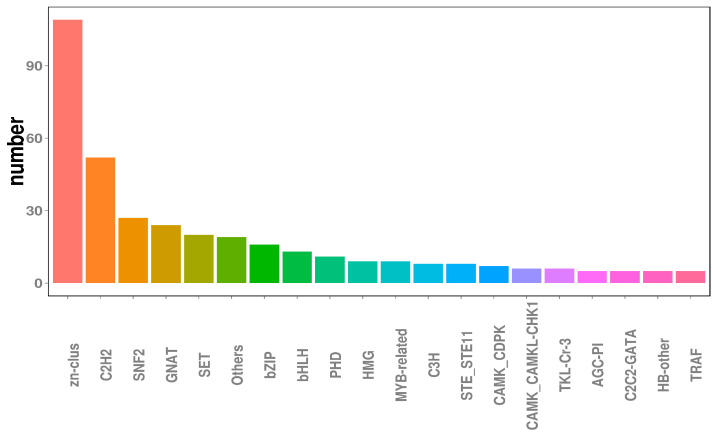
Transcription factors analysis in the *Nigrospora oryzae* genome.

**Figure 4 jof-10-00100-f004:**
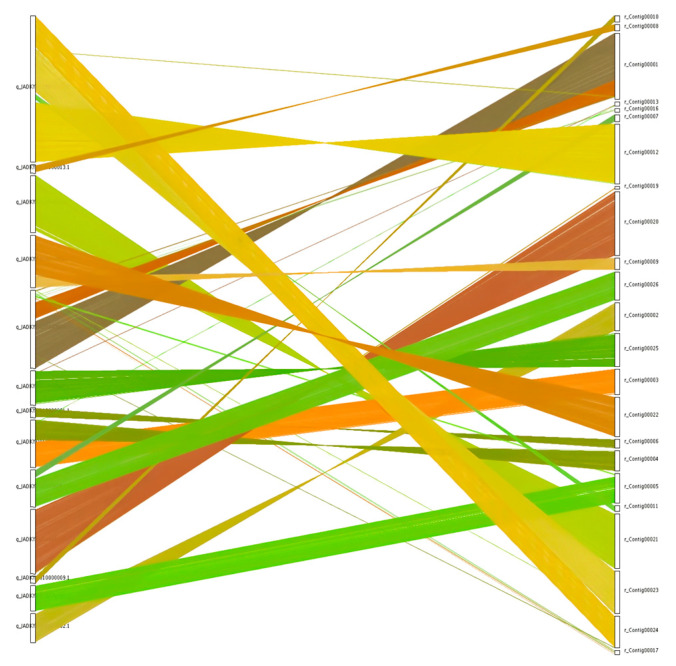
Collinearity of genome between GZL1 and ZQ1. Comparisons of ZQ1 protein sequences with the reference genome GZL1 were made through BLAST analysis. Nucleic acid level crosstalk between the genomes pairs was then obtained based on the position of the homologous genes on the genome sequence with different colour and plotted using MCScanX.

**Table 1 jof-10-00100-t001:** The basic characteristics of *Nigrospora oryzae*.

Features	*N* *. oryzae*
Assembly size (Mb)	42.09
Scaffolds	26
GC (%)	58.83
Repeated sequences (%)	2.93
Protein-coding genes	10,688
Gene density (genes per Mb)	254
Secreted proteins	879
tRNA	272
Pseudogenes	17
Average gene length (bp)	2356

**Table 2 jof-10-00100-t002:** Gene family classification statistics of GZL1 and ZQ1.

Species	Total Gene Number	Cluster Gene Number	Total Gene Family Number	Unique Gene Family Number
GZL1	10,544	10,028	9588	27
ZQ1	10,688	10,068	9585	24

## Data Availability

Data are contained within the article and Appendix A.

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
