# Peer review of "Genome Sequencing and Analysis of Nigrospora oryzae, a Rice Leaf Disease Fungus"

_jof, 2024, doi:10.3390/jof10020100_

Round 1

Reviewer 1 Report

Comments and Suggestions for Authors

The authors sequenced and annotated the genome of two strains of an emerging phytopathogen. The work is sound and adequately described.

Minor comments:

- always italicize scientific species names

- line 99- ", n ground" to ", ground"

- sections 1.2 and 1.3 have the same title

- not consistent with using commas or not for digit grouping (ex "1000" vs "1,000")

- need more spacing between length ticks (more than 5 kb) in Fig 1

- line 199- remove space between "https" and "://"

- Fig 2 text cut off, please increase text size for readers with vision difficulties

- Fig 5- just organize one side so that it's readily apparent that the genomes are collinear. The first impression was that Fig 4 must be wrong

- Fig S5 missing in figure folder, but present in supplementary Word file

Comments on the Quality of English Language

none

Author Response

Thank you very much for your careful review and meaningful suggestions. We have made each reply in the attachment. The revised manuscript will be uploaded later.

Reviewer 2 Report

Comments and Suggestions for Authors

Comments on the Quality of English Language

Author Response

Thanks for your constructive comments, we have revised the manuscript.

Reviewer 3 Report

Comments and Suggestions for Authors

This manuscript is devoted to Nigrospora oryzae, a fungal species which causes severe diseases of crop plant and other plant species worldwide. The authors in the Introduction very well explained the occurrence and importance of this fungal species. In the current work, the authors presented a genome assembly for a strain of N. oryzae isolated from a diseased rice plant in China. This manuscript should be published in JoF. However, some adjustments should be made first (see Remarks). Among other things, attention is drawn to the poor Discussion.

Remarks

Line 26 rather 1. Introduction (change the numbers of subsequent chapters )

Line 27 Nigrospora oryzae – it should be in italic, as in title

Line 29 Nigrospora (formerly Nosporia) – this needs to be checked

Line 29 Nigrospora, Oryza sativa (Line 30), Hibiscus mutabilis (line 32), Chrysanthemum morifolium (Line 66), Bacillus siamensis (line 68) - it should be in italic. Appropriate changes should be made to the entire manuscript.

Line 87 from the field - the place of collection should be precisely specified, because the place of collection and the date of isolation of the fungus should be provided when characterizing fungal strain

Line 87 How many plants were the isolations made from, how many strains/isolates were grown. How the ZQ 1 strain was selected and where it came from and how the reference strain GZL1 was selected

Line 176 Fig. – it should rather be Figure, as in other places, it should be unified

Line 177 initially white to ashen in color – how does this differ from the data in Line 171 (it is unclear why this information is provided twice)

Line 178 - It is still unclear how the strains were selected for further testing

Line 192 zq-1 – rather ZQ 1

Figure 2 caption does not correspond to the data in Figure 2

Figure 5 – requires more detailed explanation

Discussion - requires supplementation, the final fragment, where the Results are repeated and Figures are cited, should be changed. It would be necessary to comment on selected aspects of the results obtained in relation to other fungi, and to pay attention to the significance of the results obtained.

Literature - you need to write the proper names of plants, bacteria, fungi in italic and correct numerous minor errors

Line 350 (B. and Br.) fetch - it needs correction

Line 351 -it should be Oryza instead of oryza

Line 353 chrysanthemum – write with a capital letter

Line 354 sScreening - it needs correction

Line 358 Cubense – it has to be in lower case

Line 385 editorial board ??

Author Response

(The authors gave the same response as above.)

Round 2

Reviewer 2 Report

Comments and Suggestions for Authors

I thank the reviewers for addressing most of the comments I provided after th e first revision. 

I think that they may have misunderstood my comment regarding confirming the species identification. There is a section in the manuscript that confirms that the specific isolate used was correct, using growth, morphology, ITS sequencing and BLAST analysis. 

Typically, BLAST is not sufficient for species identification, especially when only using ITS, and usually, phylogenetic methods are also used. However, when combined with growth rate and morphology- they most likely have the correct species.

However, it is also essential to confirm that the ITS sequence amplified from the culture (or any gene region amplified from the isolate prior to genome sequencing) matches the ITS sequence in the genome exactly. There are many instances of genome sequences being contaminated with bacteria and other fungi due to contaminated cultures and/or contaminated DNA ahead of sequencing. 

Thus, in addition to confirming the species of the particular isolate used, you also need to confirm the "identity" of the genome.  

Author Response

Thank you very much for your suggestions and opinions on our manuscript. While revising the manuscript, we improved the quality of the manuscript and learned useful professional knowledge. The reviewer also patiently explained the methods and ideas of fungal identification in the manuscript, thank you again.